# Auto Seg-Loss: Searching Metric Surrogates for Semantic Segmentation

**Hao Li**[1*†]**, Chenxin Tao**[2*†]**, Xizhou Zhu**[3]**, Xiaogang Wang**[1,3]**, Gao Huang**[2]**, Jifeng Dai**[3,4‡]
[1]The Chinese University of Hong Kong      [2]Tsinghua University
[3]SenseTime Research      [4]Qing Yuan Research Institute, Shanghai Jiao Tong University
`haoli@link.cuhk.edu.hk, tcx20@mails.tsinghua.edu.cn`
`{zhuwalter, daijifeng}@sensetime.com`
`xgwang@ee.cuhk.edu.hk, gaohuang@tsinghua.edu.cn`

## Abstract

Designing proper loss functions is essential in training deep networks. Especially in the field of semantic segmentation, various evaluation metrics have been proposed for diverse scenarios. Despite the success of the widely adopted cross-entropy loss and its variants, the mis-alignment between the loss functions and evaluation metrics degrades the network performance. Meanwhile, manually designing loss functions for each specific metric requires expertise and significant manpower. In this paper, we propose to automate the design of metric-specific loss functions by searching differentiable surrogate losses for each metric. We substitute the non-differentiable operations in the metrics with parameterized functions, and conduct parameter search to optimize the shape of loss surfaces. Two constraints are introduced to regularize the search space and make the search efficient. Extensive experiments on PASCAL VOC and Cityscapes demonstrate that the searched surrogate losses outperform the manually designed loss functions consistently. The searched losses can generalize well to other datasets and networks. Code shall be released at `https://github.com/fundamentalvision/Auto-Seg-Loss`.

## 1 Introduction

Loss functions are of indispensable components in training deep networks, as they drive the feature learning process for various applications with specific evaluation metrics. However, most metrics, like the commonly used 0-1 classification error, are non-differentiable in their original forms and cannot be directly optimized via gradient-based methods. Empirically, the cross-entropy loss serves well as an effective surrogate objective function for a variety of tasks concerning categorization. This phenomenon is especially prevailing in image semantic segmentation, where various evaluation metrics have been designed to address the diverse task focusing on different scenarios. Some metrics measure the accuracy on the whole image, while others focus more on the segmentation boundaries. Although cross-entropy and its variants work well for many metrics, the mis-alignment between network training and evaluation still exist and inevitably leads to performance degradation.

Typically, there are two ways for designing metric-specific loss functions in semantic segmentation. The first is to modify the standard cross-entropy loss to meet the target metric (Ronneberger et al., 2015; Wu et al., 2016). The other is to design other clever surrogate losses for specific evaluation metrics (Rahman & Wang, 2016; Milletari et al., 2016). Despite the improvements, these handcrafted losses need expertise and are non-trivial to extend to other evaluation metrics.

In contrast to designing loss functions manually, an alternative approach is to find a framework that can design proper loss functions for different evaluation metrics in an automated manner, motivated by recent progress in AutoML (Zoph & Le, 2017; Pham et al., 2018; Liu et al., 2018; Li et al., 2019). Although automating the design process for loss functions is attractive, it is non-trivial to apply an

---

[*]Equal contribution. [†]This work is done when Hao Li and Chenxin Tao are interns at SenseTime Research.
[‡]Corresponding author.

AutoML framework to loss functions. Typical AutoML algorithms require a proper search space, in which some search algorithms are conducted. Previous search spaces are either unsuitable for loss design, or too general to be searched efficiently. Recently Li et al. (2019) and Wang et al. (2020) proposed search spaces based on existing handcrafted loss functions. And the algorithm searches for the best combination. However, these search spaces are still limited to the variants of cross-entropy loss, and thus do not address the mis-alignment problem well.

In this paper, we propose a general framework for searching surrogate losses for mainstream non-differentiable segmentation metrics. The key idea is that we can build the search space according to the form of evaluation metrics. In this way, the training criteria and evaluation metrics are unified. Meanwhile, the search space is compact enough for efficient search. Specifically, the metrics are first relaxed to the continuous domain by substituting the one-hot prediction and logical operations, which are the non-differentiable parts in most metrics, with their differentiable approximations. Parameterized functions are introduced to approximate the logical operations, ensuring that the loss surfaces are smooth while effective for training. The loss parameterization functions can be of arbitrary families defined on $[0, 1]$. Parameter search is further conducted on the chosen family so as to optimize the network performance on the validation set with the given evaluation metric. Two essential constraints are introduced to regularize the parameter search space. We find that the searched surrogate losses can effectively generalize to different networks and datasets. Extensive experiments on Pascal VOC (Everingham et al., 2015) and Cityscapes (Cordts et al., 2016) show our approach delivers accuracy superior than the existing losses specifically designed for individual segmentation metrics with a mild computational overhead.

Our contributions can be summarized as follows: 1) Our approach is the first general framework of surrogate loss search for mainstream segmentation metrics. 2) We propose an effective parameter regularization and parameter search algorithm, which can find loss surrogates optimizing the target metric performance with mild computational overhead. 3) The surrogate losses obtained via the proposed searching framework promote our understandings on loss function design and by themselves are novel contributions, because they are different from existing loss functions specifically designed for individual metrics, and are transferable across different datasets and networks.

## 2 RELATED WORK

**Loss function design** is an active topic in deep network training (Ma, 2020). In the area of image semantic segmentation, cross-entropy loss is widely used (Ronneberger et al., 2015; Chen et al., 2018). But the cross-entropy loss is designed for optimizing the global accuracy measure (Rahman & Wang, 2016; Patel et al., 2020), which is not aligned with many other metrics. Numerous studies are conducted to design proper loss functions for the prevalent evaluation metrics. For the mIoU metric, many works (Ronneberger et al., 2015; Wu et al., 2016) incorporate class frequency to mitigate the class imbalance problem. For the boundary F1 score, the losses at boundary regions are up-weighted (Caliva et al., 2019; Qin et al., 2019), so as to deliver more accurate boundaries. These works carefully analyze the property of specific evaluation metrics, and design the loss functions in a fully handcrafted way, which needs expertise. By contrast, we propose a unified framework for deriving parameterized surrogate losses for various evaluation metrics. Wherein, the parameters are searched by reinforcement learning in an automatic way. The networks trained with the searched surrogate losses deliver accuracy on par or even superior than those with the best handcrafted losses.

**Direct loss optimization** for non-differentiable evaluation metrics has long been studied for structural SVM models (Joachims, 2005; Yue et al., 2007; Ranjbar et al., 2012). However, the gradients w.r.t. features cannot be derived from these approaches. Therefore, they cannot drive the training of deep networks through back-propagation. Hazan et al. (2010) proposes to optimize structural SVM with gradient descent, where loss-augmented inference is applied to get the gradients of the expectation of evaluation metrics. Song et al. (2016) further extends this approach to non-linear models (e.g., deep neural networks). However, the computational complexity is very high during each step in gradient descent. Although Song et al. (2016) and Mohapatra et al. (2018) have designed efficient algorithms for the Average Precision (AP) metric, other metrics still need specially designed efficient algorithms. Our method, by contrast, is general for the mainstream segmentation metrics. Thanks to the good generalizability, our method only needs to perform the search process

once for a specific metric, and the searched surrogate loss can be directly used henceforth. Applying the searched loss for training networks brings very little additional computational cost.

**Surrogate loss** is introduced to derive loss gradients for the non-differentiable evaluation metrics. There are usually two ways for designing surrogate losses. The first is to handcraft an approximated differentiable metric function. For the IoU measure, Rahman & Wang (2016) propose to approximate the intersection and union seperately using the softmax probabilities in a differentiable form, and show its effectiveness on binary segmentation tasks. Berman et al. (2018) further deal with multi-class segmentation problems by extending mIoU from binary inputs to the continuous domain with the convex Lovàsz extension, and their method outperforms standard cross entropy loss in multi-class segmentation tasks. For the F1 measure, dice loss is proposed by Milletari et al. (2016) as a direct objective by substituting the binary prediction with the softmax probability. In spite of the success, they do not apply for other metrics.

The second solution is to train a network to approximate the target metric. Nagendar et al. (2018) train a network to approximate mIoU. Patel et al. (2020) design a neural network to learn embeddings for predictions and ground truths for tasks other than segmentation. This line of research focuses on minimizing the approximation error w.r.t. the target metrics. But there is no guarantee that their approximations provide good loss signals for training. These approximated losses are just employed in a post-tuning setup, still relying on cross-entropy pre-trained models. Our method significantly differs in that we search surrogate losses to directly optimize the evaluation metrics in applications.

**AutoML** is a long-pursued target of machine learning (He et al., 2019). Recently a sub-field of AutoML, neural architecture search (NAS), has attracted much attention due to its success in automating the process of neural network architecture design (Zoph & Le, 2017; Pham et al., 2018; Liu et al., 2018). As an essential element, loss function has also raised the interest of researchers to automate its design process. Li et al. (2019) and Wang et al. (2020) design search spaces based on existing human-designed loss functions and search for the best combination parameters. There are two issues: a) the search process outputs whole network models rather than loss functions. For every new network or dataset, the expensive search procedure is conducted again, and b) the search space are filled with variants of cross-entropy, which cannot solve the mis-alignment between cross-entropy loss and many target metrics. By contrast, our method outputs the searched surrogate loss functions of close form with the target metrics, which are transferable between networks and datasets.

## 3 REVISITING EVALUATION METRICS FOR SEMANTIC SEGMENTATION

Various evaluation metrics are defined for semantic segmentation, to address the diverse task focusing on different scenarios. Most of them are of three typical classes: Acc-based, IoU-based, and F1-score-based. This section revisits the evaluation metrics, under a unified notation set.

Table 1 summarizes the mainstream evaluation metrics. The notations are as follows: suppose the validation set is composed of $N$ images, labeled with categories from $C$ classes (background included). Let $I_n, n \in \{1, \ldots, N\}$ be the $n$-th image, and $Y_n$ be the corresponding ground-truth segmentation mask. Here $Y_n = \{y_{n,c,h,w}\}_{c,h,w}$ is a one-hot vector, where $y_{n,c,h,w} \in \{0, 1\}$ indicates whether the pixel at spatial location $(h, w)$ belongs to the $c$-th category ($c \in \{1, \ldots, C\}$). In evaluation, the ground-truth segmentation mask $Y_n$ is compared to the network prediction $\hat{Y}_n = \{\hat{y}_{n,c,h,w}\}_{c,h,w}$, where $\hat{y}_{n,c,h,w} \in \{0, 1\}$. $\hat{y}_{n,c,h,w}$ is quantized from the continuous scores produced by the network (by argmax operation).

**Acc-based metrics.** The global accuracy measure (gAcc) counts the number of pixels correctly classified. It can be written with logical operator AND as Eq. (1). The gAcc metric counts each pixel equally, so the results of the long-tailed categories have little impact on the metric number. The mean accuracy (mAcc) metric mitigates this by normalizing within each category as in Eq. (2).

**IoU-based metrics.** The evaluation is on set similarity rather than pixel accuracy. The intersection-over-union (IoU) score is evaluated between the prediction and the ground-truth mask of each category. The mean IoU (mIoU) metric averages the IoU scores of all categories, as in Eq. (3).

In the variants, the frequency weighted IoU (FWIoU) metric weighs each category IoU score by the category pixel number, as in Eq. (4). The boundary IoU (BIoU) (Kohli et al., 2009) metric only cares about the segmentation quality around the boundary, so it picks the boundary pixels out in evaluation

Table 1: Revisiting mainstream metrics for semantic segmentation. The metrics with † measure the segmentation accuracy on the whole image. The metrics with ∗ focus on the boundary quality.

| Type | Name | Formula | |
|---|---|---|---|
| Acc-based | Global Accuracy† | $\text{gAcc} = \dfrac{\sum_{n,c,h,w} \hat{y}_{n,c,h,w} \text{ AND } y_{n,c,h,w}}{\sum_{n,c,h,w} y_{n,c,h,w}}$ | (1) |
| | Mean Accuracy† | $\text{mAcc} = \dfrac{1}{C} \sum_c \dfrac{\sum_{n,h,w} \hat{y}_{n,c,h,w} \text{ AND } y_{n,c,h,w}}{\sum_{n,h,w} y_{n,c,h,w}}$ | (2) |
| IoU-based | Mean IoU† | $\text{mIoU} = \dfrac{1}{C} \sum_c \dfrac{\sum_{n,h,w} \hat{y}_{n,c,h,w} \text{ AND } y_{n,c,h,w}}{\sum_{n,h,w} \hat{y}_{n,c,h,w} \text{ OR } y_{n,c,h,w}}$ | (3) |
| | Frequency Weighted IoU† | $\text{FWIoU} = \sum_c \dfrac{\sum_{n,h,w} y_{n,c,h,w}}{\sum_{n,c',h,w} y_{n,c',h,w}} \dfrac{\sum_{n,h,w} \hat{y}_{n,c,h,w} \text{ AND } y_{n,c,h,w}}{\sum_{n,h,w} \hat{y}_{n,c,h,w} \text{ OR } y_{n,c,h,w}}$ | (4) |
| | Boundary IoU∗ | $\text{BIoU} = \dfrac{1}{C} \sum_c \dfrac{\sum_n \sum_{h,w \in \text{BD}(y_n)} \hat{y}_{n,c,h,w} \text{ AND } y_{n,c,h,w}}{\sum_n \sum_{h,w \in \text{BD}(y_n)} \hat{y}_{n,c,h,w} \text{ OR } y_{n,c,h,w}}$
where $\text{BD}(y) = y \text{ XOR Min-Pooling}(y)$ | (5) |
| F1-score-based | Boundary F1 Score∗ | $\text{BF1-score} = \dfrac{1}{C} \sum_c \dfrac{2 \times \text{prec}_c \times \text{recall}_c}{(\text{prec}_c + \text{recall}_c)}$
where $\text{prec}_c = \dfrac{\sum_{n,h,w} \text{BD}(\hat{y}_n)_{c,h,w} \text{ AND Max-Pooling}(\text{BD}(y_n)_{c,h,w})}{\sum_{n,h,w} \text{BD}(\hat{y}_n)_{c,h,w}}$,
$\text{recall}_c = \dfrac{\sum_{n,h,w} \text{Max-Pooling}(\text{BD}(\hat{y}_n)_{c,h,w}) \text{ AND}(\text{BD}(y_n)_{c,h,w})}{\sum_{n,h,w} \text{BD}(y_n)_{c,h,w}}$ | (6) |

and ignores the rest pixels. It can be calculated with Eq. (5), in which $\text{BD}(y_n)$ denotes the boundary region in map $y_n$. $\text{BD}(y_n)$ is derived by applying XOR operation on the min-pooled ground-truth mask. The stride of the $\text{Min-Pooling}(\cdot)$ is 1.

**F1-score-based metrics.** F1-score is a criterion that takes both precision and recall into consideration. A well-known metric of this type is boundary F1-score (BF1-score) (Csurka et al., 2013), which is widely used for evaluating boundary segmentation accuracy. The computation of precision and recall in BF1-score is as in Eq. (6), where $\text{BD}(\hat{y}_n)$ and $\text{BD}(y_n)$ are derived from Eq. (5). Max pooling with stride 1, $\text{Max-Pooling}(\cdot)$, is applied on the boundary regions to allow error tolerance.

# 4 AUTO SEG-LOSS FRAMEWORK

In the Auto Seg-Loss framework, the evaluation metrics are transferred into continuous surrogate losses with learnable parameters, which are further optimized. Fig. 1 illustrates our approach.

## 4.1 EXTENDING METRICS TO SURROGATES

As shown in Section 3, most segmentation metrics are non-differentiable because they take one-hot prediction maps as input, and contain binary logical operations. We extend these metrics to be continuous loss surrogates by smoothing the non-differentiable operations within.

**Extending One-hot Operation.** The one-hot prediction map, $\hat{Y}_n = \{\hat{y}_{n,c,h,w}\}_{c,h,w}$, is derived by picking the highest scoring category at each pixel, which is further turned into one-hot form. Here, we approximate the one-hot predictions with softmax probabilities, as,

$$\hat{y}_{n,c,h,w} \approx \widetilde{y}_{n,c,h,w} = \text{Softmax}_c\left(z_{n,c,h,w}\right), \tag{7}$$

where $z_{n,c,h,w} \in \mathbb{R}$ is the category score output by the network (without normalization). The approximated one-hot prediction is denoted by $\widetilde{y}_{n,c,h,w}$.

**Extending Logical Operations.** As shown in Table 1, the non-differentiable logical operations, $f_{\text{AND}}(y_1, y_2)$, $f_{\text{OR}}(y_1, y_2)$, and $f_{\text{XOR}}(y_1, y_2)$, are of indispensable components in these metrics. Because the XOR operation can be constructed by AND and OR, $f_{\text{XOR}}(y_1, y_2) = f_{\text{OR}}(y_1, y_2) - f_{\text{AND}}(y_1, y_2)$, we focus on extending $f_{\text{AND}}(y_1, y_2)$ and $f_{\text{OR}}(y_1, y_2)$ to the continuous domain.

Following the common practice, the logical operators are substituted with arithmetic operators

$$f_{\text{AND}}(y_1, y_2) = y_1 y_2, \quad f_{\text{OR}}(y_1, y_2) = y_1 + y_2 - y_1 y_2, \tag{8}$$

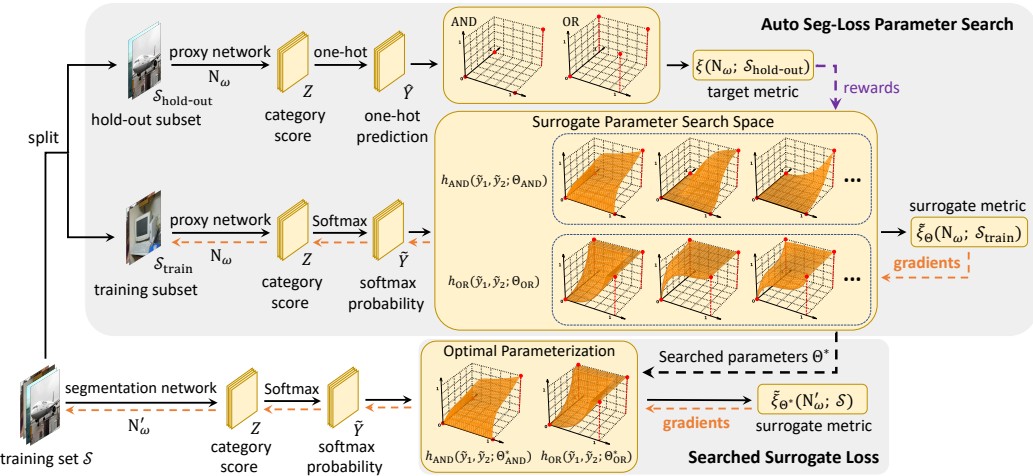

Figure 1: Overview of the proposed Auto Seg-Loss framework. The surfaces of $h_{\mathrm{AND}}$ and $h_{\mathrm{OR}}$ shown in the "Optimal Parameterization" illustrate the searched optimal parameterization for mIoU.

where $y_1, y_2 \in \{0, 1\}$. Eq. (8) can be directly extended to take continuous $y_1, y_2 \in [0, 1]$ as inputs. By such an extension, together with the approximated one-hot operation, a naïve version of differentiable surrogate losses can be obtained. The strength of such surrogates is that they are directly derived from the metrics, which significantly reduces the gap between training and evaluation. However, there is no guarantee that the loss surfaces formed by naïvely extending Eq. (8) provide accurate loss signals. To adjust the loss surfaces, we parameterize the AND and OR functions as

$$
\begin{aligned}
h_{\mathrm{AND}}(y_1, y_2; \theta_{\mathrm{AND}}) &= g(y_1; \theta_{\mathrm{AND}})\, g(y_2; \theta_{\mathrm{AND}}), \\
h_{\mathrm{OR}}(y_1, y_2; \theta_{\mathrm{OR}}) &= g(y_1; \theta_{\mathrm{OR}}) + g(y_2; \theta_{\mathrm{OR}}) - g(y_1; \theta_{\mathrm{OR}})\, g(y_2; \theta_{\mathrm{OR}}),
\end{aligned} \tag{9}
$$

where $g(y; \theta) : [0, 1] \to \mathbb{R}$ is a scalar function parameterized by $\theta$.

The parameterized function $g(y; \theta)$ can be from arbitrary function families defined on $[0, 1]$, e.g., piecewise linear functions and piecewise Bézier curves. With a chosen function family, the parameters $\theta$ control the shape of loss surfaces. We seek to search for the optimal parameters $\theta$ so as to maximize the given evaluation metric.

Meanwhile, optimal parameter search is non-trivial. With the introduced parameters, the plasticity of loss surfaces is strong. The parameterized loss surfaces may well be chaotic, or be far away from the target evaluation metric even at the binary inputs. For more effective parameter search, we regularize the loss surfaces by introducing two constraints on $g(y; \theta)$.

*Truth-table constraint* is introduced to enforce the surrogate loss surfaces taking the same values as the evaluation metric score at binary inputs. This is applied by enforcing

$$
g(0; \theta) = 0, \ g(1; \theta) = 1. \tag{10}
$$

Thus, the parameterized functions $h(y_1, y_2; \theta)$ preserve the behavior of the corresponding logical operations $f(y_1, y_2)$ on binary inputs $y_1, y_2 \in \{0, 1\}$.

*Monotonicity constraint* is introduced based on the observation of monotonicity tendency in the truth tables of AND and OR. It pushes the loss surfaces towards a benign landscape, avoiding dramatic non-smoothness. The monotonicity constraint is enforced on $h_{\mathrm{AND}}(y_1, y_2)$ and $h_{\mathrm{OR}}(y_1, y_2)$, as

$$
\partial h_{\mathrm{AND}}/\partial y_i \geq 0, \ \partial h_{\mathrm{OR}}/\partial y_i \geq 0, \ \forall y_i \in [0, 1], \ i = 1, 2.
$$

Applying the chain rule and the truth table constraint, the monotonicity constraint implies

$$
\partial g(y; \theta)/\partial y \geq 0, \ \forall y \in [0, 1]. \tag{11}
$$

Empirically we find it important to enforce these two constraints in parameterization.

**Extending Evaluation Metrics.** Now we can extend the metrics to surrogate losses by a) replacing the one-hot predictions with softmax probabilities, and b) substituting the logical operations with parameterized functions. Note that if the metric contains several logical operations, their parameters will not be shared. The collection of parameters in one metric are denoted as $\Theta$. For a segmentation network N and evaluation dataset $\mathcal{S}$, the score of the evaluation metric is denoted as $\xi(\mathrm{N}; \mathcal{S})$. And the parameterized surrogate loss is denoted as $\widetilde{\xi}_\Theta(\mathrm{N}; \mathcal{S})$.

## 4.2 SURROGATE PARAMETERIZATION

The parameterized function can be from any function families defined on [0, 1], such as picewise Bézier curve and piecewise linear functions. Here we choose the piecewise Bézier curve for parameterizing $g(y; \theta)$, which is widely used in computer graphics and is easy to enforce the constraints via its control points. We also verify the effectiveness of parameterizing $g(y; \theta)$ by piecewise linear functions. See Fig. 2 for visualization and Appendix B for more details.

A piecewise Bézier curve consists of a series of quadratic Bézier curves, where the last control point of one curve segment coincides with the first control point of the next curve segment. If there are $n$ segments in a piecewise Bézier curve, the $k$-th segment is defined as

$$B(k, s) = (1 - s)^2 B_{2k} + 2s(1 - s)B_{2k+1} + s^2 B_{2k+2}, \ 0 \le s \le 1 \tag{12}$$

where $s$ transverses the $k$-th segment, $B_{2k+i} = (B_{(2k+i),u}, B_{(2k+i),v})$ $(i = 0, 1, 2)$ denotes the $i$-th control point on the $k$-th segment, in which $u, v$ index the 2-d plane axes. A piecewise Bézier curve with $n$ segments has $2n + 1$ control points in total. To parameterize $g(y; \theta)$, we assign

$$y = (1 - s)^2 B_{2k,u} + 2s(1 - s)B_{(2k+1),u} + s^2 B_{(2k+2),u}, \tag{13a}$$
$$g(y; \theta) = (1 - s)^2 B_{2k,v} + 2s(1 - s)B_{(2k+1),v} + s^2 B_{(2k+2),v}, \tag{13b}$$
$$\text{s.t. } B_{2k,u} \le y \le B_{(2k+2),u}, \tag{13c}$$

where $\theta$ is the control point set, $B_{2k,u} < B_{(2k+1),u} < B_{(2k+2),u}, 0 \le k \le n - 1$. Given an input $y$, the segment index $k$ and the transversal parameter $s$ are derived from Eq. (13c) and Eq. (13a), respectively. Then $g(y; \theta)$ is assigned as Eq. (13b). Because $g(y; \theta)$ is defined on $y \in [0, 1]$, we arrange the control points in the $u$-axis as, $B_{0,u} = 0$, $B_{2n,u} = 1$, where the $u$-coordinate of the first and the last control points are at $0$ and $1$, respectively.

The strength of the piecewise Bézier curve is that the curve shape is defined explicitly via the control points. Here we enforce the truth-table and the monotonicity constraints on the control points via,

$$B_{0,v} = 0, \ B_{2n,v} = 1; \qquad \qquad \textit{(truth-table constraint)}$$
$$B_{2k,v} \le B_{(2k+1),v} \le B_{(2k+2),v}, \quad k = 0, 1, \ldots, n - 1. \qquad \textit{(monotonicity constraint)}$$

To fulfill the above restrictions in optimization, the specific form of the parameters is given by

$$\theta = \left\{ \left( \frac{B_{i,u} - B_{(i-1),u}}{B_{2n,u} - B_{(i-1),u}}, \frac{B_{i,v} - B_{(i-1),v}}{B_{2n,v} - B_{(i-1),v}} \right) \mid i = 1, 2, \ldots, 2n - 1 \right\},$$

with $B_0 = (0, 0)$ and $B_{2n} = (1, 1)$ fixed. So every $\theta_i = (\theta_{i,u}, \theta_{i,v})$ is in range $[0, 1]^2$ and it is straight-forward to compute the actual coordinates of control points from this parameterized form. Such parameterization makes each $\theta_i$ independent with each other, and thus simplifies the optimization. By default, we use piecewise Bézier curve with two segments to parameterize $g(y, \theta)$.

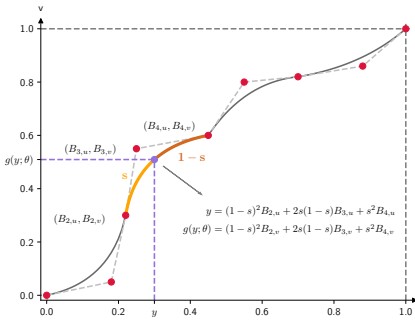

Figure 2: Parameterization of $g(y; \theta)$ using Piecewise Bézier curve with four segments. The red points are control points. The purple point is on the curve, which shows the relationship among $y$, $g(y; \theta)$ and the transversal parameter $s$.

---

**Algorithm 1:** Auto Seg-Loss Parameter Search

**Input:** Initialized network $N_{\omega_0}$, initialized distribution $\mu_1$ and $\sigma^2$, target metric $\xi$, training set $\mathcal{S}_{\text{train}}$ and hold-out training set $\mathcal{S}_{\text{hold-out}}$

**Result:** Obtained optimal parameters $\Theta^*$

**for** $t = 1$ to $T$ **do**
 **for** $i = 1$ to $M$ **do**
  Sample parameter $\Theta_i^{(t)} \sim \mathcal{N}_{\text{trunc}[0,1]}(\mu_t, \sigma^2 I)$;
  Network training
   $\omega^*(\Theta_i^{(t)}) = \arg\max_\omega \widetilde{\xi}_{\Theta_i^{(t)}}(N_\omega; \mathcal{S}_{\text{train}})$,
  with $w$ initialized from $w_0$;
  Compute the evaluation metric score
   $\xi(\Theta_i^{(t)}) = \xi(N_{\omega^*(\Theta_i^{(t)})}; \mathcal{S}_{\text{hold-out}})$;
 **end**
 Update $\mu_{t+1} = \arg\max_\mu \frac{1}{M} \sum_{i=1}^M R(\mu, \mu_t, \Theta_i^{(t)})$;
**end**
**return** $\Theta^* = \arg\max_{\mu_t} \sum_{i=1}^M \xi(\Theta_i^{(t)}), \forall t = 1, \ldots, T + 1$

### 4.3 Surrogate parameter optimization

Algorithm 1 describes our parameter search algorithm. The training set is split into two subsets, $\mathcal{S}_{\text{train}}$ for training and $\mathcal{S}_{\text{hold-out}}$ for evaluation in the search algorithm, respectively. Specifically, suppose we have a segmentation network $N_\omega$ with weights $\omega$, our search target is the parameters that maximize the evaluation metric on the hold-out training set $\xi(N_\omega; \mathcal{S}_{\text{hold-out}})$

$$\max_\Theta \xi(\Theta) = \xi(N_{\omega^*(\Theta)}; \mathcal{S}_{\text{hold-out}}), \quad \text{s.t.} \quad \omega^*(\Theta) = \arg\max_\omega \widetilde{\xi}_\Theta(N_\omega; \mathcal{S}_{\text{train}}). \quad (14)$$

To optimize Eq. (14), the segmentation network is trained with SGD as the inner-level problem. At the outer-level, we use reinforcement learning as our searching algorithm, following the common practice in AutoML (Zoph & Le, 2017; Pham et al., 2018). Other searching algorithms, such as evolutionary algorithm, may also be employed. Specifically, the surrogate parameters are searched via the PPO2 algorithm (Schulman et al., 2017). The process consists of $T$ sampling steps. In the $t$-th step, we aim to explore the search space around that from $t - 1$. Here $M$ sets of parameters $\{\Theta_i^{(t)}\}_{i=1}^M$ are sampled independently from a truncated normal distribution (Burkardt, 2014), as $\Theta \sim \mathcal{N}_{\text{trunc}[0,1]}(\mu_t, \sigma^2 I)$, with each variable in range $[0, 1]$. In it, $\mu_t$ and $\sigma^2 I$ denote the mean and covariance of the parent normal distribution ($\sigma$ is fixed as 0.2 in this paper). $\mu_t$ summarizes the information from the $(t - 1)$-th step. $M$ surrogate losses are constructed with the sampled parameters, which drive the training of $M$ segmentation networks separately. To optimize the outer-level problem, we evaluate these models with the target metric and take the evaluation scores as rewards for PPO2. Following the PPO2 algorithm, $\mu_{t+1}$ is computed as $\mu_{t+1} = \arg\max_\mu \frac{1}{M} \sum_{i=1}^M R(\mu, \mu_t, \Theta_i)$, where the reward $R(\mu, \mu_t, \Theta_i)$ is as

$$R(\mu, \mu_t, \Theta_i) = \min\left( \frac{p(\Theta_i; \mu, \sigma^2 I)}{p(\Theta_i; \mu_t, \sigma^2 I)} \xi(\Theta_i), \text{ CLIP}\left( \frac{p(\Theta_i; \mu, \sigma^2 I)}{p(\Theta_i; \mu_t, \sigma^2 I)}, 1 - \epsilon, 1 + \epsilon \right) \xi(\Theta_i) \right),$$

where $\min(\cdot, \cdot)$ picks the smaller item from its inputs, $\text{CLIP}(x, 1 - \epsilon, 1 + \epsilon)$ clips $x$ to be within $1 - \epsilon$ and $1 + \epsilon$, and $p(\Theta_i; \mu, \sigma^2 I)$ is the PDF of the truncated normal distribution. Note that the mean reward of the $M$ samples is subtracted when computing $\xi(\Theta_i)$ for better convergence. After $T$ steps, the mean $\mu_t$ with the highest average evaluation score is output as the final parameters $\Theta^*$.

Empirically we find the searched losses have good transferability, i.e., they can be applied for different datasets and networks. Benefiting from this, we use a light proxy task for parameter search. In it, we utilize a smaller image size, a shorter learning schedule and a lightweight network. Thus, the whole search process is quite efficient (8 hours on PASCAL VOC with 8 NVIDIA Tesla V100 GPUs). More details are in Appendix A. In addition, the search process can be conducted only once for a specific metric and the resulting surrogate loss can be directly used for training henceforth.

## 5 Experiments

We evaluate on the PASCAL VOC 2012 (Everingham et al., 2015) and the Cityscapes (Cordts et al., 2016) datasets. We use Deeplabv3+ (Chen et al., 2018) with ResNet-50/101 (He et al., 2016) as the network model. During the surrogate parameter search, we randomly sample 1500 training images in PASCAL VOC and 500 training images in Cityscapes to form the hold-out set $\mathcal{S}_{\text{hold-out}}$, respectively. The remaining training images form the training set $\mathcal{S}_{\text{train}}$ in search. $\mu_0$ is set to make $g(y; \theta) = y$. The backbone network is ResNet-50. The images are down-sampled to be of $128 \times 128$ resolution. SGD lasts only 1000 iterations with a mini-batch size of 32. After the search procedure, we re-train the segmentation networks with ResNet-101 using the searched losses on the full training set and evaluate them on the actual validation set. The re-train settings are the same as Deeplabv3+ (Chen et al., 2018), except that the loss function is substituted by the obtained surrogate loss. The search time is counted on 8 NVIDIA Tesla V100 GPUs. More details are in Appendix A.

### 5.1 Searching for Different Metrics

In Table 2, we compare our searched surrogate losses against the widely-used cross-entropy loss and its variants, and some other metric-specific surrogate losses. We also seek to compare with the AutoML-based method in Li et al. (2019), which was originally designed for other tasks. But we cannot get reasonable results due to convergence issues. The results show that our searched losses

are on par or better the previous losses on their target metrics. It is interesting to note that the obtained surrogates for boundary metrics (such as BIoU and BF1) only focus on the boundary areas, see Appendix C for further discussion. We also tried training segmentation networks driven by both searched mIoU and BIoU/BF1 surrogate losses. Such combined losses refine the boundaries while keeping reasonable global performance.

Table 2: Performance of different losses on PASCAL VOC and Cityscapes segmentation. The results of each loss function's target metrics are underlined. The scores whose difference with the highest is less than 0.3 are marked in **bold**.

| Dataset | PASCAL VOC | | | | | | Cityscapes | | | | | |
|---|---|---|---|---|---|---|---|---|---|---|---|---|
| Loss Function | mIoU | FWIoU | BIoU | BF1 | mAcc | gAcc | mIoU | FWIoU | BIoU | BF1 | mAcc | gAcc |
| Cross Entropy | 78.69 | 91.31 | 70.61 | 65.30 | 87.31 | 95.17 | 79.97 | **93.33** | 62.07 | 62.24 | 87.01 | **96.44** |
| WCE (Ronneberger et al., 2015) | 69.60 | 85.64 | 61.80 | 37.59 | **92.61** | 91.11 | 73.01 | 90.51 | 53.07 | 51.19 | **89.22** | 94.56 |
| DPCE (Caliva et al., 2019) | 79.82 | 91.76 | 71.87 | 66.54 | 87.76 | 95.45 | 80.27 | 93.38 | 62.57 | 65.99 | 86.99 | 96.46 |
| SSIM (Qin et al., 2019) | 79.26 | 91.68 | 71.54 | 66.35 | 87.87 | 95.38 | **80.65** | 93.22 | 63.04 | 72.20 | 86.88 | 96.39 |
| DiceLoss (Milletari et al., 2016) | 77.78 | 91.34 | 69.85 | 64.38 | 87.47 | 95.11 | 79.30 | 93.25 | 60.93 | 59.94 | 86.38 | 96.39 |
| Lovàsz (Berman et al., 2018) | 79.72 | 91.78 | 72.47 | 66.65 | 88.64 | 95.42 | 77.67 | 92.51 | 56.71 | 53.48 | 82.05 | 96.03 |
| Searched mIoU | **80.97** | 92.09 | 73.44 | 68.86 | 88.23 | 95.68 | **80.67** | 93.30 | 63.05 | 67.97 | 87.20 | 96.44 |
| Searched FWIoU | 80.00 | **91.93** | 75.14 | 65.67 | 89.23 | 95.44 | 79.42 | **93.33** | 61.71 | 59.68 | 87.96 | 96.37 |
| Searched BIoU | 48.97 | 69.89 | **79.27** | 38.99 | 81.28 | 62.64 | 45.89 | 39.80 | **63.89** | 38.29 | 62.80 | 58.15 |
| Searched BF1 | 1.93 | 0.96 | 7.39 | **74.83** | 6.51 | 2.66 | 6.78 | 3.19 | 18.37 | **77.40** | 12.09 | 8.19 |
| Searched mAcc | 69.80 | 85.86 | 72.85 | 35.62 | **92.66** | 91.28 | 74.10 | 90.79 | 54.62 | 53.45 | **89.22** | 94.75 |
| Searched gAcc | 79.73 | 91.76 | 74.09 | 64.41 | 88.95 | **95.47** | 79.41 | 93.30 | 61.65 | 62.04 | 87.08 | **96.51** |
| Searched mIoU + BIoU | **81.19** | 92.19 | 76.89 | 69.56 | 88.36 | 95.75 | **80.43** | 93.34 | **63.88** | 65.87 | 87.03 | 96.45 |
| Searched mIoU + BF1 | 78.72 | 90.80 | 71.81 | 73.57 | 86.70 | 94.88 | 78.30 | 93.00 | 61.62 | 71.73 | 87.13 | **96.23** |

## 5.2 GENERALIZATION OF THE LOSS

**Generalization among datasets.** Table 3 evaluates the generalization ability of our searched loss surrogates among different datasets. Due to limited computational resource, we train networks only with the searched mIoU, BF1 and mAcc surrogate losses. The results show that our searched surrogate losses generalize well between these two datasets with quite different scenes and categories.

Table 3: Generalization of our searched surrogate losses between PASCAL VOC and Cityscapes.

| Datasets | Cityscapes ⟶ VOC | | | | | | VOC ⟶ Cityscapes | | | | | |
|---|---|---|---|---|---|---|---|---|---|---|---|---|
| Loss Function | mIoU | FWIoU | BIoU | BF1 | mAcc | gAcc | mIoU | FWIoU | BIoU | BF1 | mAcc | gAcc |
| Cross Entropy | 78.69 | 91.31 | 70.61 | 65.30 | 87.31 | **95.17** | 79.97 | **93.33** | 62.07 | 62.24 | 87.01 | **96.44** |
| Searched mIoU | **80.05** | **91.72** | **73.97** | 67.61 | 88.01 | **95.45** | **80.67** | 93.31 | 62.96 | 66.48 | 87.36 | **96.44** |
| Searched BF1 | 1.84 | 0.93 | 7.42 | **75.85** | 6.48 | 1.47 | 6.67 | 3.20 | 19.00 | **77.99** | 12.12 | 4.09 |
| Searched mAcc | 70.90 | 86.29 | 73.43 | 37.18 | **93.19** | 91.43 | 73.50 | 90.68 | 54.34 | 54.04 | **88.66** | 94.68 |

**Generalization among segmentation networks.** The surrogate losses are searched with ResNet-50 + DeepLabv3+ on PASCAL VOC. The searched losses drive the training of ResNet-101 + DeepLabv3+, PSPNet (Zhao et al., 2017) and HRNet (Sun et al., 2019) on PASCAL VOC. Table 4 shows the results. The results demonstrate that our searched loss functions can be applied to various semantic segmentation networks.

## 5.3 ABLATION

**Parameterization and constraints.** Table 5 ablates the parameterization and the search space constraints. In it, a surrogate without parameters refers to Eq. (8), with the domain extended from discrete points $\{0, 1\}$ to continuous interval $[0, 1]$. This naive surrogate deliver much lower accuracy, indicating the essence of parameterization. Without the truth-table constraint, the training process diverges at the very beginning, where the loss gradients become "NaN". And the performance drops if the monotonicity constraint is not enforced. The performance drops or even the algorithm fails without the constraints.

Table 4: Generalization of our searched surrogate losses among different network architectures on PASCAL VOC. The losses are searched with ResNet-50 + DeepLabv3+ on PASCAL VOC.

| Network | R50-DeepLabv3+ | | | R101-DeepLabv3+ | | | R101-PSPNet | | | HRNetV2p-W48 | | |
|---|---|---|---|---|---|---|---|---|---|---|---|---|
| Loss Function | mIoU | BF1 | mAcc | mIoU | BF1 | mAcc | mIoU | BF1 | mAcc | mIoU | BF1 | mAcc |
| Cross Entropy | 76.22 | 61.75 | 85.43 | 78.69 | 65.30 | 87.31 | 77.91 | 64.70 | 85.71 | 76.35 | 61.19 | 85.12 |
| Searched mIoU | **78.35** | 66.93 | 85.53 | **80.97** | 68.86 | 88.23 | **78.93** | 65.65 | 87.42 | **77.26** | 63.52 | 86.80 |
| Searched BF1 | 1.35 | **70.81** | 6.05 | 1.93 | **74.83** | 6.51 | 1.62 | **71.84** | 6.33 | 1.34 | **68.41** | 5.99 |
| Searched mAcc | 69.82 | 36.92 | **91.61** | 69.80 | 35.62 | **92.66** | 71.66 | 39.44 | **92.06** | 68.22 | 35.90 | **91.46** |

**Proxy tasks for parameter search.** Table 6 ablates this. The bottom row is our default setting with a light-weight backbone, down-sampled image size and shorter learning schedule. The default setting delivers on par accuracy with heavier settings. This is consistent with the generalization ability of our surrogate losses. Thus we can improve the search efficiency via light proxy tasks.

**Parameter search algorithm.** Fig. 3 compares the employed PPO2 (Schulman et al., 2017) algorithm with random search. The much better performance of PPO2 suggests that surrogate loss search is non-trivial and reinforcement learning helps to improve the search efficiency.

Table 5: Ablation on search space constraints.

| Parameter | Truth-table | Monotonicity | VOC mIoU |
|---|---|---|---|
| ✗ | ✗ | ✗ | 46.99 |
| ✓ | ✗ | ✗ | Fail |
| ✓ | ✓ | ✗ | 77.76 |
| ✓ | ✓ | ✓ | 80.64 |

Table 6: Ablation on search proxy tasks.

| Backbone | Image Size | Iterations | Time(hours) | VOC mIoU |
|---|---|---|---|---|
| R50 | $256 \times 256$ | 1000 | 33.0 | 81.15 |
| R50 | $128 \times 128$ | 2000 | 17.1 | 80.56 |
| R101 | $128 \times 128$ | 1000 | 13.3 | 80.75 |
| R50 | $128 \times 128$ | 1000 | 8.5 | 80.97 |

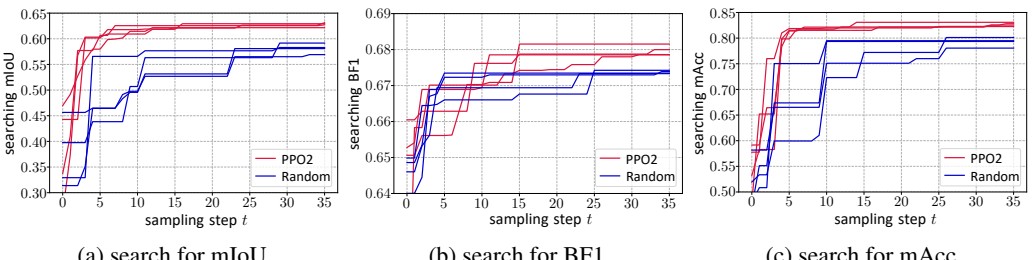

(a) search for mIoU    (b) search for BF1    (c) search for mAcc

Figure 3: Ablation on loss parameter search. Each curve presents the highest average evaluation score up to the $t$-th step in one search process. The search process is repeated four times.

## 6 CONCLUSION

The introduced Auto Seg-Loss is a powerful framework to search for the parameterized surrogate losses for mainstream segmentation evalutation metrics. The non-differentiable operators are substituted by their parameterized continuous counterparts. The parameters are optimized to improve the final evaluation metrics with essential constraints. It would be interesting to extend the framework to more tasks, like object detection, pose estimation and machine translation problems.

ACKNOWLEDGMENTS

The work is supported by the National Key R&D Program of China (2020AAA0105200), Beijing Academy of Artificial Intelligence and the Institute for Guo Qiang of Tsinghua University.

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
