# OpenReview forum: "Auto Seg-Loss: Searching Metric Surrogates for Semantic Segmentation"
_ICLR.cc/2021/Conference — ICLR 2021 Poster_

### Official Review · AnonReviewer4 · 2020-10-27

**Rating:** 7
**Confidence:** 4

**Review:**

In this paper, the authors suggest using differentiable surrogate parameterized loss functions that more closely approximate some of the frequently used metrics for segmentation, including variations of accuracy, IoU, and F-score for the whole area and the boundaries, and use reinforcement learning to tune the parameters instantiating the surrogate loss functions.
Moderate improvement is shown through experimental evaluations compared to cross-entropy and other used loss functions.

I think, overall, this is a good paper, the following elaborates it:

Strengths:
* I like the general idea of bridging the gap between the often-non-differentiable evaluation metrics and the optimization loss functions so that the model is more closely optimizing for what the task mostly cares about.
* Despite the existence of several previous works on the taken general direction, there are still seemingly novel methodological contributions including the loss parameterization and its tuning process. The design of most of the components made intuitive sense.
* The experimental setup including comparisons to different losses, ablation, and transferability studies are thorough. A moderate but consistent improvement is observable when compared to the frequently used loss functions, over two commonly used semantic segmentation datasets, and across various network architectures.
* The paper is fairly well-written and easy to follow.

Weak points:
* Even though designing differentiable surrogate losses is conceptually simple and straight-forward, as shown in the experimental section, in practice it turns out that a relatively complicated set of tricks is needed to be leveraged before it shows improvements, including the parameterization and its two regularization schemes and the parameterization tuning using reinforcement learning. This in total adds significant complexity on top of a normal training scheme.
* Some of the other distance-based metrics including the Hausdorff and Chamfer distance used to evaluate semantic segmentation have not been covered in this work.

Minor comments:
* In table 4, for R101-DeepLabv3+, R101-PSPNet and HRNetV2p-W48, I am missing comparisons to their ceiling, i.e. the values they could have obtained if the parameters were directly optimized for them.
* Algorithm 1: \mu_0, for t=1... and (\mu_t, ...) seem not consistent. Please fix.
* Equation (4) variable c is ambiguous in the left denominator as being the variable for both nested sigmas, please change the inner, e.g. to c'.
* Equation (2) is not computationally stable as the denominator can turn to 0, in case a class c is not present in the dataset.
* In Table 5, "Fail" under VOC mIoU is ambiguous; in case a very low performance is obtained, I think reporting the number would still help.
* In the text of section 3, IoU-based metrics, please specify that Min-Pooling and Max-Pooling are with stride 1.
* Typo: "the our searched loss" => "that our searched loss"

---

> ### Author Response · Authors · 2020-11-23
> **Response to AnonReviewer4**
>
> We clarify the questions as follows.
>
> ---
> Q1: Even though designing differentiable surrogate losses is conceptually simple and straight-forward, as shown in the experimental section, in practice it turns out that a relatively complicated set of tricks is needed to be leveraged before it shows improvements, including the parameterization and its two regularization schemes and the parameterization tuning using reinforcement learning. This in total adds significant complexity on top of a normal training scheme.
>
> A1: First of all, we would like to emphasize that these techniques are not tricks, because they are essential in applying AutoML to semantic segmentation loss function design. Moreover, they also provide useful guidance for loss function search in other tasks. We consider them part of our key contributions and highlight their importance in the introduction part. In the main body of the paper, we carefully described the motivations and the key insights in designing them. They are quite different from those "complicated set of tricks", which lack insights and motivations.
>
> As for the computational complexity, we emphasize that the search step is conducted only once when dealing with a new evaluation metric, and the searched loss can be directly used henceforth. We discussed the time consumption in Table 6. The total search time for mIoU is 8.5 hours with 8 NVIDIA Tesla V100 GPUs. We note that the search time is quite efficient as an AutoML-based method. Applying the searched loss for training new networks brings very little extra computational cost.
>
> ---
> Q2: Some of the other distance-based metrics including the Hausdorff and Chamfer distance used to evaluate semantic segmentation have not been covered in this work.
>
> A2: To the best of our knowledge, the Hausdorff and Chamfer distances are rarely used in the field of semantic segmentation. These metrics are mainly used for medical image segmentation, which is not the main task of this paper. Moreover, these metrics cannot be directly used on the PASCAL VOC and Cityscapes datasets because they are defined on the binary segmentation task. Actually, in the paper, we have covered the mainstream evaluation metrics for semantic segmentation as in [1,2,3,4,5].
>
> On the other hand, our framework can also be generalized to the Hausdorff and Chamfer distances, which is illustrated in the next comment due to the limited characters.
>
> [1] Csurka et al., What Is A Good Evaluation Measure for Semantic Segmentation?, BMVC 2013.
> [2] Long et al., Fully Convolutional Networks for Semantic Segmentation, CVPR 2015.
> [3] Zhao et al., Pyramid Scene Parsing Network, CVPR 2017.
> [4] Wang et al., Deep High-Resolution Representation Learning for Visual Recognition, TPAMI, 2020.
> [5] Chen et al., Encoder-Decoder with Atrous Separable Convolution for Semantic Image Segmentation, ECCV 2018.
>
> ---
> Q3: In table 4, for R101-DeepLabv3+, R101-PSPNet and HRNetV2p-W48, I am missing comparisons to their ceiling, i.e. the values they could have obtained if the parameters were directly optimized for them.
>
> A3:Thanks for the suggestion. We explored the ceiling for R101-DeepLabv3+, which can be found in the third line of Table 6 (VOC mIoU 80.75). Because it does not show improvement over the generalized setting (search with R50-Deeplabv3+), we did not conduct this experiment on R101-PSPNet and HRNetV2p-W48 further.
>
> ---
> Q4: Equation (2) is not computationally stable as the denominator can turn to 0, in case a class c is not present in the dataset.
>
> A4: To make the formulations simple and clear, here we assume that all the categories are contained in the dataset without loss of generality. It’s true that the denominator will be 0 if a class is not present in the dataset, but this situation will not happen in our experiments.
>
> ---
> Q5: In Table 5, "Fail" under VOC mIoU is ambiguous; in case a very low performance is obtained, I think reporting the number would still help.
>
> A5: "Fail" means that the search algorithm diverges at the very beginning, where the loss gradients become "NaN". In the uploaded revision, we discussed this.
>
> ---
> Q6: Other typos.
>
> A6: Thanks for the careful reading and suggestion. We fixed them in the uploaded revision.

---

> > ### Author Response · Authors · 2020-11-23
> > **Generalization to the Hausdorff and Chamfer distances**
> >
> > Our framework can also be generalized to the Hausdorff and Chamfer distances as follows:
> >
> > (1) The Hausdorff distance is defined as $\mathrm{HD}(\hat{Y}, Y) = \max⁡(\mathrm{hd}(\hat{Y}, Y), \mathrm{hd}(Y, \hat{Y}))$ , where $\hat{Y} \in \mathbb{R}^{H\times W}$ and $Y  \in \mathbb{R}^{H\times W}$ denotes the prediction and target, respectively. $\mathrm{hd}(\hat{Y}, Y) = \max_{\hat{y}\in \mathrm{BD}(\hat{Y})}\mathrm{⁡min}_{y\in \mathrm{BD}(Y)}||\hat{y}-y||_2$ , and $\mathrm{hd}(Y, \hat{Y})$ is defined by swapping $\hat{Y}$ and $Y$.
> >
> > Following [6], the Hausdorff distance can be rewritten as $\mathrm{HD}(\hat{Y}, Y) = \max(\max_{\hat{y}\in \hat{Y}}((\hat{Y}\ \mathrm{XOR}\ Y) \circ d_Y), \max_{y\in Y}((Y\ \mathrm{XOR}\ \hat{Y}) \circ d_{\hat{Y}}))$ , where $\circ$ denotes the elementwise multiplication, and $d_Y \in \mathbb{R}^{H\times W}$ denotes the distance transform map, in which each element denotes the minimum distance from $\hat{y}$ to the boundary $\mathrm{BD}(Y)$. In calculating $d_{\hat{Y}}$, we do not substitute $\hat{Y}$ with corresponding logits. Note that the only non-differentiable component in this formula is the XOR term, thus our framework can be applied to the Hausdorff distance.
> >
> > (2) The Chamfer distance is defined as $\mathrm{CH}(\hat{Y}, Y) = \mathrm{ch}(\hat{Y}, Y) + \mathrm{ch}(Y, \hat{Y})$ , where $\mathrm{ch}(\hat{Y}, Y) = \sum_{\hat{y}\in \mathrm{BD}(\hat{Y})}\mathrm{⁡min}_{y\in \mathrm{BD}(Y)}||\hat{y}-y||_2$ , and $\mathrm{ch}(Y, \hat{Y})$ is defined by swapping $\hat{Y}$ and $Y$.
> >
> > Similarly, the function $\mathrm{CH}(\hat{Y}, Y)$ can be rewritten as $\mathrm{CH}(\hat{Y}, Y) = \sum_{\hat{y}\in \hat{Y}}(\mathrm{BD}(\hat{Y})\ \mathrm{AND}\ (\hat{Y}\ \mathrm{XOR}\ Y) \circ d_Y) + \sum_{y\in Y}(\mathrm{BD}(Y)\ \mathrm{AND}\ (Y\ \mathrm{XOR}\ \hat{Y}) \circ d_{\hat{Y}})$. The boundary can be calculated via $\mathrm{BD}(\hat{Y}) = \hat{Y}\  \mathrm{XOR}\ \mathrm{MinPooling}(\hat{Y})$, which follows the practice in Eq.(5). Compared with the Hausdorff distance, the differences are that the Chamfer distance computes the sum of selected distance transform values instead of the max of that, and a boundary mask is used for selecting boundary pixels. Thus, the non-differentiable component in this formula is the AND and XOR term, and our framework can be applied to the Chamfer distance.
> >
> > [6] Karimi & Salcudean, Reducing The Hausdorff Distance in Medical Image Segmentation with Convolutional Neural Networks, IEEE T-MI, 2019.

---

### Official Review · AnonReviewer3 · 2020-10-28
**In this work, the authors proposed an Auto Seg-Loss architecture for semantic segmentation. The authors first extended the traditional metrics to surrogates, by replacing the one-hot operation with the softmax one, and the non-differentiable logical operations with the arithmetic ones. Next, the authors employed the piecewise Bezier curve for surrogate parameterization.**

**Rating:** 5
**Confidence:** 3

**Review:**

Major strengths:

1.) In comparison with traditional loss function such as Cross-Entropy, WCE, DPCE, and SSIM, the proposed method achieves competitive performance. In addition, the authors also compare with the searched loss functions such as searched mIoU, searched FWIoU, etc. By combining the searched mIoU with the BIoU/BFI surrogate losses, the overall method achieves reasonable global performance, while refines the boundaries.

2.) Generalization experiments are conducted to demonstrate that the proposed surrogate losses are effectiveness under various scenes and categories.

3.) The overall paper is clearly presented and easy to follow.


Major weaknesses:

1.) Lack of computational cost analysis. According to the implementational details, the overall training process has two steps, which is likely to increase the computational burden. To this end, the authors are suggested to conduct some analysis on this issue.

2.) The comparison experiments with some recent and important methods are missing. For example, the following three papers also focus on Auto-ML based semantic segmentation:

[1] FasterSeg: Searching for Faster Real-time Semantic Segmentation, ICLR 2020;
[2] Graph-guided Architecture Search for Real-time Semantic Segmentation, CVPR 2020;
[3] Auto-DeepLab: Hierarchical Neural Architecture Search for Semantic Image Segmentation, CVPR 2019.

As the Auto Seg-Loss is particularly designed for Auto-ML based semantic segmentation, the comparison with the related methods is required.

In addition, there are some other recent semantic segmentation methods (proposed in 2019 and 2020) focusing on the PASCAL VOC and Cityscapes datasets, although they are not related to Auto-ML. As an effective semantic segmentation method, the overall segmentation performance of the model is supposed to be competitive. To this end, the authors are also suggested to make comparisons with these SOTA semantic segmentation methods.

3.) The writing of the abstract needs to be improved. In the current manuscript, the abstract is too brief. To better attract the interests of the readers, the authors can first introduce the background of the problem studied in this paper. Next, some detailed motivations can also be included.

---

> ### Author Response · Authors · 2020-11-23
> **Response to AnonReviewer3**
>
> Thanks for the constructive comments. We clarify the questions as follows.
>
> ---
> Q1:
> Lack of computational cost analysis. According to the implementational details, the overall training process has two steps, which is likely to increase the computational burden. To this end, the authors are suggested to conduct some analysis on this issue.
>
> A1:
> We discussed the time consumption in Table 6. The total search time for mIoU is 8.5 hours with 8 NVIDIA Tesla V100 GPUs. We note that the search time is quite efficient as an AutoML-based method.
>
> We further emphasize that the search process only needs to be conducted once for a specific metric, and the searched surrogate loss can be directly used henceforth. Applying the searched loss for training networks brings very little additional computational cost.
>
> ---
> Q2:
> The comparison experiments with some AutoML-based and SOTA semantic segmentation methods are missing:
> 1)The authors should compare their method with other Auto-ML based semantic segmentation methods such as [1,2,3], because Auto Seg-Loss is particularly designed for Auto-ML based semantic segmentation;
> 2)The authors should also make comparisons with recent SOTA semantic segmentation methods, since the overall performance is supposed to be competitive for an effective semantic segmentation method.
>
> [1] Chen et al., FasterSeg: Searching for Faster Real-time Semantic Segmentation, ICLR 2020.
> [2] Lin et al., Graph-guided Architecture Search for Real-time Semantic Segmentation, CVPR 2020.
> [3] Liu et al., Auto-DeepLab: Hierarchical Neural Architecture Search for Semantic Image Segmentation, CVPR 2019.
>
> A2:
> Our paper focuses on the design of loss functions, which is orthogonal to the network architecture improvements in the network architecture search papers[1,2,3]. Specifically, these papers all focus on searching network architectures while just adopt normal cross-entropy based losses for training. By contrast, our method focuses on searching for loss functions while keeping the network unchanged. Therefore, our method is not directly comparable to these papers.
>
> With regard to the SOTA in semantic segmentation, we choose Deeplabv3+ for our main experiments because it is widely used and provides nearly SOTA results. Specifically, current SOTA result on the Cityscapes validation set(82.7) is achieved by combining Deeplabv3+ with a much stronger backbone ResNeSt200[4], while Deeplabv3+ with Resnet101 as backbone (used as the baseline in our paper) can deliver 79.97 mIoU. The improvement only comes from the stronger backbone. We have verified the transferability between different network architectures of our searched surrogate losses in Table 4.
>
> [4] Zhang et al., ResNeSt: Split-Attention Networks, arXiv preprint, 2020.
>
> ---
> Q3:
> The writing of the abstract needs to be improved. In the current manuscript, the abstract is too brief. To better attract the interests of the readers, the authors can first introduce the background of the problem studied in this paper. Next, some detailed motivations can also be included.
>
> A3:
> Thanks for the suggestion. The current "too brief" problem is mainly due to limited space. In the uploaded revision, we update the abstract. Specifically, we first introduce the problem background to point out the gap between loss functions and evaluation metrics. Then the motivations and search framework are summarized. Finally the experimental results are displayed.

---

> > ### Author Response · Authors · 2020-11-23
> > **Modification of abstract**
> >
> > We paste the updated abstract as follows:
> >
> > "Designing proper loss functions is essential in training deep networks. Especially in the field of semantic segmentation, various evaluation metrics have been proposed for diverse scenarios. Despite the success of the widely adopted cross-entropy loss and its variants, the mis-alignment between the loss functions and evaluation metrics degrades the network performance. Meanwhile, manually designing loss functions for each specific metric requires expertise and significant manpower. In this paper, we propose to automate the design of metric-specific loss functions by searching differentiable surrogate losses for each metric. We substitute the non-differentiable operations in the metrics with parameterized functions, and conduct parameter search to optimize the shape of loss surfaces. Two constraints are introduced to regularize the search space and make the search efficient. Extensive experiments on PASCAL VOC and Cityscapes demonstrate that the searched surrogate losses outperform the manually designed loss functions consistently. The searched losses can generalize well to other datasets and networks. Code shall be released."

---

### Official Review · AnonReviewer2 · 2020-10-28
**The formulation is interesting, but the paper is poorly written**

**Rating:** 5
**Confidence:** 3

**Review:**

This paper aims to directly optimize the metrics of semantic segmentation tasks, such as mIoU, which is different from the most existing methods which minimize the cross-entropy as a proxy. The metrics typically contain one-hot labels and logical operations. In order to directly optimize them, the authors first relax the one-hot label/prediction by Softmax. Then the logical operations applied on the one-hot label are extended by a continuous parameter function which is Monotonical and has the same output as the logical operation with 0/1 input. Finally, the authors describe a reinforcement learning framework to optimize the metrics parameterization (i.e., the outer objective), while the inner objective (i.e., the segmentation network) is trained by standard SGD. The experiments have been performed on Pascal VOC 2012 and Cityscapes datasets, showing the searched loss outperformed the traditional ones such as cross-entropy.

Pros:
1. The Bezier curves parameterization is used to guarantee a smooth relaxation, and two additional constraints are proposed to regularize the parameterization.
2. Two-stage optimization is proposed to learn the network weights and the loss parameterization alternatively.
3. The ablations on the generalization-ability is conducted.

Cons:
1. My main concern is that this paper is poorly written and difficult to understand. The abstract and the introduction do not provide necessary information about the high-level design of the proposed algorithm.
2. The method is also not well motivated. For example:
2a. Why using reinforcement learning to optimize the outer objective? Is there any other option?
2b. Are there any other alternatives to the piecewise Bezier curves? Is the algorithm sensitive to a different number of Bezier segments?
3. Some necessary technical details are missing, for example, what does "No Parameters" mean in Table 5? Does it mean a single straight line from (0, 0) to (1, 1)?

In summary, this paper may contain interesting idea and algorithm implementation, I encourage the authors to further polish their presentation to better express them.

---

> ### Author Response · Authors · 2020-11-23
> **Response to AnonReviewer2**
>
> Thanks for the constructive comments. We admit that some parts of the paper are too brief and not clear enough, mainly due to the limited space. We tried very hard to shorten the paper into 8 pages before submission. Now with one additional page, we carefully revised the paper and provided many more details. Especially, we provide more illustrations for high-level design in abstract and introduction. We hope R#2 can check the revision and find it satisfactory.
>
> ---
> Q1: My main concern is that this paper is poorly written and difficult to understand. The abstract and the introduction do not provide necessary information about the high-level design of the proposed algorithm.
>
> A1: We admit that we did not emphasize our motivation enough due to the limited space. In revision, we improved our abstract by first introducing the problem background, pointing out the gap between loss functions and evaluation metrics. Then we summarized the motivations and search framework. In Introduction, we explained our motivation of utilizing AutoML for loss function design in detail.
>
> ---
> Q2: Why using reinforcement learning to optimize the outer objective? Is there any other option?
>
> A2: We use RL as our search algorithm for the outer-level optimization following the common practice in AutoML [1,2]. Other options, like evolutionary algorithm and particle swarm optimization, may well work well within our framework. We emphasize this in Section 4.3 of the revised version.
>
> [1] Barret Zoph and Quoc V. Le, Neural architecture search with reinforcement learning, ICLR 2017.
> [2] Hieu Pham, Melody Guan, Barret Zoph, Quoc Le, and Jeff Dean, Efficient neural architectu research via parameters sharing, ICML 2018.
>
> ---
> Q3: Are there any other alternatives to the piecewise Bezier curves? Is the algorithm sensitive to a different number of Bezier segments?
>
> A3:
> 1) As stated in the first paragraph of section 4.2, we also tried piecewise linear functions and verified its effectiveness in Appendix B. We emphasize this more in the revision.
> 2) As for the number of Bezier segments, we found using more than 3 Bezier segments did not improve the final performance noticeably while increasing the parameters. we also tried one-segment Bezier curves, which performs slightly worse than two-segment ones. So we use piecewise Bezier curves with 2 segments as our default setting.
>
> ---
> Q4: Some necessary technical details are missing, for example, what does "No Parameters" mean in Table 5? Does it mean a single straight line from (0, 0) to (1, 1)?
>
> A4: We mentioned this in Section 4.1 under the Equation (8). The "naïve surrogate without parameters" refers to the extended Equation (8), with its domain extended naïvely from discrete points {0, 1} to continuous interval [0, 1]. In revision, we update the discussion part of Table 5 to further emphasize this.

---

> > ### Author Response · Authors · 2020-11-23
> > **Modification of abstract and introduction**
> >
> > The abstract is updated as follows:
> >
> > "Designing proper loss functions is essential in training deep networks. Especially in the field of semantic segmentation, various evaluation metrics have been proposed for diverse scenarios. Despite the success of the widely adopted cross-entropy loss and its variants, the mis-alignment between the loss functions and evaluation metrics degrades the network performance. Meanwhile, manually designing loss functions for each specific metric requires expertise and significant manpower. In this paper, we propose to automate the design of metric-specific loss functions by searching differentiable surrogate losses for each metric. We substitute the non-differentiable operations in the metrics with parameterized functions, and conduct parameter search to optimize the shape of loss surfaces. Two constraints are introduced to regularize the search space and make the search efficient. Extensive experiments on PASCAL VOC and Cityscapes demonstrate that the searched surrogate losses outperform the manually designed loss functions consistently. The searched losses can generalize well to other datasets and networks. Code shall be released."
> >
> > To further illustrate our motivation, we add the following paragraphs to introduction:
> >
> > "In contrast to designing loss functions manually, an alternative approach is to find a framework that can design proper loss functions for different evaluation metrics in an automated manner, motivated by recent progress in AutoML (Zoph & Le, 2017; Pham et al., 2018; Liu et al., 2018; Li et al., 2019). Although automating the design process for loss functions is attractive, it is non-trivial to apply an AutoML framework to loss functions. Typical AutoML algorithms require a proper search space, in which some search algorithms are conducted. Previous search spaces are either unsuitable for loss design, or too general to be searched efficiently. Recently Li et al. (2019) and Wang et al. (2020) propose search spaces based on existing handcrafted loss functions. And the algorithm searches for the best combination. However, these search spaces are still limited to the variants of cross-entropy loss, and thus do not address the mis-alignment problem well.
> >
> > In this paper, we propose a general framework for searching surrogate losses for mainstream non-differentiable segmentation metrics. The key idea is that we can build the search space according to the form of evaluation metrics. In this way, the training criteria and evaluation metrics are unified. Meanwhile, the search space is compact enough for efficient search."

---

### Official Review · AnonReviewer5 · 2020-11-06
**Careful study of semantic segmentation proxy loss functions**

**Rating:** 7
**Confidence:** 3

**Review:**

This paper looks at loss functions for semantic segmentation. Typical metrics are not easily differentiable w.r.t. the outputs of the DNNs that generate the labels. Instead, a proxy/surrogate loss function is learned jointly with the network in a two-level optimization.

Pros:

i) Good accuracy results. Code will allow others to verify and build on these.

ii) High-quality ablation studies.

Avoiding any parameterization (one of the the more surprising components of the method) is a good ablation to have. Also, comparison to random in Fig 3 demonstrates that the outer-level optimization is working as expected.

Q1: What is the naive surrogate used in Table 5? Equation (8)?

Cons:

iii) Seems like an ad-hoc approach that doesn't incorporate classical techniques for optimizing "hard" loss functions.

Non-differentiable loss functions, or losses over discrete variables, are less common when dealing with CNNs. But they can be included in deep learning with techniques like that in:

Chen et. al. "Learning Deep Structured Models"

Notable citations that take this a "classical" approach are:

Ranjbar et. al., "Optimizing Non-Decomposable Loss Functions in Structured Prediction"

This kind of approach is an omission from related work that might be worth correcting. Especially because it is the "direct" approach, in that it is optimizing the original loss functions using known techniques, without additional levels of learning or approximatinos as in the submission.

Overall, though the empirical results are strong enough that it is likely that the submission is doing something useful well. So the approach in the submission seems well-supported by that fact, even if previous literature makes it a non-obvious way to do things.

iv) Not much illustration or exploration of the learned loss surfaces.

Experiments partially leave open the question: How closely is the surrogate loss matching the target metrics? Basically, is there more direct evidence that for the intuition that h_AND is approximately equivalent to f_AND? The networks trained on the surrogate losses do well on the original metrics. But it's not impossible that some trivial solution or unexpected loss function, that is not clearly similar to e.g. mIoU, can train a network to learn a segmentation model that produces segmentations with good mIoU.

Minor comments:

Q2: Reason for bolding in Table 2 is unclear, what is a "(co)-highest result?"

---

> ### Author Response · Authors · 2020-11-23
> **Response to AnonReviewer5**
>
> Thanks for the constructive comments. We clarify the questions as follows.
>
> ---
> Q1: What is the naïve surrogate used in Table 5? Is it Equation (8)?
>
> A1: Yes. The naïve surrogate refers to the extended Equation (8), with its domain extended naïvely from discrete points {0, 1} to continuous interval [0, 1]. We mentioned this in Section 4.1 under the Equation (8).  In revision, we updated the discussion part of Table 5 for clarification.
>
> ---
> Q2: Some previous literature that aims to optimize the "hard" metrics is not included:
> [1] Chen et. al., "Learning Deep Structured Models", ICML 2015.
> [2] Ranjbar et. al., "Optimizing Non-Decomposable Loss Functions in Structured Prediction", TPAMI, 2013.
> And Auto Seg-Loss does not incorporate the techniques in these papers.
>
> A2: Thanks for the suggestion. The paper [1] is irrelevant with our work, because it is designed for maximizing the data likelihood (equivalent to maximizing the cross-entropy), and cannot be extended for arbitrary evaluation metrics. On the other hand, it uses CNNs to learn the potential functions for MRF models, which is not an usual paradigm for deep semantic segmentation.
>
> The paper [2] indeed has the same goal of directly optimizing towards the given metrics as our work. However, it is designed for the structural SVMs, where the gradients w.r.t. features cannot be derived. Therefore, it cannot drive the training of deep networks through back-propagation. We discuss the paper [2] and the related works for "direct loss optimization" in the uploaded revision.
>
> ---
> Q3: How closely is the surrogate loss matching the target metrics? Basically, is there more direct evidence that for the intuition that h_AND is approximately equivalent to f_AND?
>
> A3: Our goal is not to approximate the target metrics, but to search surrogate losses that can train the networks well under the target metrics. The h_AND in surrogate losses does not need to be "approximately equivalent to" f_AND in target metrics.
>
> On the other hand, we indeed enforce constraints on h_AND to make the search space compact. In the target metric,  f_AND is defined on discrete points {0, 1}. On these points, h_AND in the surrogate loss is enforced to take the same values as f_AND, because of the truth-table constraint.
>
> ---
> Q4: Not much illustration of the learned loss surfaces.
>
> A4: Due to the space limit, we could not give detailed visualization of the learned loss surfaces. While in the "optimal parameterization" part of Fig 1, the surfaces of h_AND and h_OR are actually the searched ones for mIoU. We shall mention that clearly in revision, with the added space.
>
> ---
> Q5: The networks trained on the surrogate losses do well on the original metrics. But it's not impossible that some trivial solution or unexpected loss function, that is not clearly similar to e.g. mIoU, can train a network to learn a segmentation model that produces segmentations with good mIoU.
>
> A5: The "unexpected" solutions may exist, but it is very difficult to find them in a systematical way. Because the space of all possible loss functions is too huge to explore. Therefore, designing suitable search space is important. By starting from the form of target metrics, our proposed search space is compact yet effective for search as shown in the experiments.
>
> There may exist possible ways to find the "unexpected" solutions, e.g. to search the loss function from basic operations or handcrafting such solutions. However, the huge search space makes searching from basic operations very computationally expensive, while handcrafting such solutions is infeasible since it requires domain expertise, repeated trials and errors, and is hard to extended to different metrics.
>
> ---
> Q6: Reason for bolding in Table 2 is unclear, what is a "(co)-highest result"?
>
> A6: The "(co)-highest results" are those results whose difference with the highest ones is less than 0.3, where the difference may well come from randomness. We will update the caption of Table 2 for better understanding. Thanks for the suggestion.

---

> > ### Author Response · Authors · 2020-11-23
> > **Modification of related work**
> >
> > We add the following paragraph to related work:
> >
> > "Direct loss optimization for non-differentiable evaluation metrics has long been studied for structural SVM models (Joachims, 2005; Yue et al., 2007; Ranjbar et al., 2012). However, the gradients w.r.t. features cannot be derived from these approaches. Therefore, they cannot drive the training of deep networks through back-propagation. Hazan et al. (2010) proposes to optimize structural SVM with gradient descent, where loss-augmented inference is applied to get the gradients of the expectation of evaluation metrics. Song et al. (2016) further extends this approach to non-linear models (e.g., deep neural networks). However, the computational complexity is very high during each step in gradient descent. Although Song et al. (2016) and Mohapatra et al. (2018) have designed efficient algorithms for the Average Precision (AP) metric, other metrics still need specially designed efficient algorithms. Our method, by contrast, is general for the mainstream segmentation metrics. Thanks to the good generalizability, our method only needs to perform the search process once for a specific metric, and the searched surrogate loss can be directly used henceforth. Applying the searched loss for training networks brings very little additional computational cost."

---

### Author Response · Authors · 2020-11-23
**Revised version of the paper**

We thank all the reviewers for their constructive comments. We have updated a revised version of the paper. In the original submission, due to the 8 page constraint, we have to make the description very compact. With the one additional page, we would like to highlight the changes as follows:

1) We provide more detailed illustrations for our motivation and high-level design in abstract and introduction.

2) We add a paragraph in the related works, discussing a series of research works (direct loss optimization).

3) We further elaborate some technical details to make them easier to understand.

---

### Decision · Program_Chairs · 2021-01-07
**Final Decision**

**Decision:**

Accept (Poster)

**Comment:**

Auto Seg-Loss uses a differentiable surrogate parameterized loss function that approximates using RL some of the non-differentiable metrics for segmentation. Auto Seg-Loss outperform cross-entropy and other loss functions through a great number of experiments. The main concerns rised by the reviewers (More clarity on the abstract and intro, extending the related work, and performance experiments) has been addressed. Accordingly I recommend the paper to be accepted at ICLR 2021.